# Strategic Advances in Combination Therapy for Metastatic Castration-Sensitive Prostate Cancer: Current Insights and Future Perspectives

**DOI:** 10.3390/cancers16183187

**Published:** 2024-09-18

**Authors:** Whi-An Kwon, Yong Sang Song, Min-Kyung Lee

**Affiliations:** 1Department of Urology, Hanyang University College of Medicine, Myongji Hospital, Goyang 10475, Republic of Korea; 2Department of Obstetrics and Gynecology, Hanyang University College of Medicine, Myongji Hospital, Goyang 10475, Republic of Korea; 3Department of Internal Medicine, Hanyang University College of Medicine, Myongji Hospital, Goyang 10475, Republic of Korea

**Keywords:** prostate cancer, metastatic castrate sensitive prostate cancer, systemic treatment, combination therapy, doublet therapy, triplet therapy

## Abstract

**Simple Summary:**

This review explores the evolution of treatment strategies for metastatic castration-sensitive prostate cancer, emphasizing the benefits of early treatment intensification with androgen deprivation therapy, androgen receptor pathway inhibitors, and chemotherapy. Despite robust evidence and guideline recommendations, their real-world adoption remains low. The review discusses clinical trial findings, real-world data, and ongoing challenges, advocating for a personalized treatment approach based on disease characteristics and patient fitness to optimize outcomes and narrow the gap between clinical evidence and practice.

**Abstract:**

The contemporary treatment for metastatic castration-sensitive prostate cancer (mCSPC) has evolved significantly, building on successes in managing metastatic castration-resistant prostate cancer (mCRPC). Although androgen deprivation therapy (ADT) alone has long been the cornerstone of mCSPC treatment, combination therapies have emerged as the new standard of care based on recent advances, offering improved survival outcomes. Landmark phase 3 trials demonstrated that adding chemotherapy (docetaxel) and androgen receptor pathway inhibitors to ADT significantly enhances overall survival, particularly for patients with high-volume, high-risk, or de novo metastatic disease. Despite these advancements, a concerning gap between evidence-based guidelines and real-world practice remains, with many patients not receiving recommended combination therapies. The challenge in optimizing therapy sequences, considering both disease control and treatment burdens, and identifying clinical and biological subgroups that could benefit from personalized treatment strategies persists. The advent of triplet therapy has shown promise in extending survival, but the uro-oncology community must narrow the gap between evidence and practice to deliver the most effective care. Current research is focused on refining treatment approaches and utilizing biomarkers to guide therapy selection, aiming to offer more personalized and adaptive strategies for mCSPC management. Thus, aligning clinical practices with the evolving evidence is urgently needed to improve outcomes for patients facing this incurable disease.

## 1. Introduction

The treatment landscape of metastatic prostate cancer has evolved tremendously over the past two decades, driven by advancements in the understanding of tumor biology, as well as the development and approval of various new agents [1]. This evolution began with the FDA approval of docetaxel for metastatic castration-resistant prostate cancer (mCRPC) in 2004. Since then, the efficacy of several other drug classes has been established, improving survival rates and setting new standards of care for patients with mCRPC [2].

A paradigm shift toward treatment intensification in the early disease stages has recently occurred. Numerous novel strategies for metastatic castration-sensitive prostate cancer (mCSPC) have arisen from therapies proven successful in mCRPC [3] (Figure 1). This shift involves using several androgen receptor pathway inhibitors (ARPIs) in addition to androgen deprivation therapy (ADT). The shift is supported by evidence demonstrating the superiority of combining ADT with novel hormonal agents over ADT monotherapy [3]. Similarly, chemotherapy, once reserved for mCRPC, has prolonged survival in selected patients with mCSPC who have high-volume (HV) metastases and have received upfront chemotherapy [4]. This evidence supports the concept of early combination therapy in patients with mCSPC.

mCSPC incidence is increasing, as evidenced by US-based studies showing the shift in the stage of prostate cancer diagnosis, which is likely influenced by changes in prostate-specific antigen (PSA) screening recommendations by the US Preventative Services Task Force [5]. While not implying direct causality, the rising incidence of metastatic prostate cancer is a high-priority issue due to the incurable nature of advanced disease, which is associated with inevitable therapy resistance and poorer survival outcomes [6]. Over the past decade, treatment for mCSPC has evolved significantly, driven by large, randomized, phase 3 clinical trials demonstrating improvements in overall survival (OS) and quality of life (QoL) with combination therapy over the historical standard of ADT alone [7].

mCSPC treatment aims to prolong survival through long-term tumor suppression by targeting tumor proliferation, particularly the androgen receptor (AR) pathway. Treatments directed at the AR pathway include ADT and ARPIs [8]. ADT reduces the amount of androgen available for AR binding, diminishes AR-mediated cell signaling, and increases cell cycle arrest and apoptosis [9]. ARPIs, such as abiraterone acetate, apalutamide, darolutamide, and enzalutamide, inhibit androgen synthesis or compete to bind to the AR, thereby disrupting the cell cycle [10]. Taxanes, such as docetaxel, stabilize microtubules, prevent mitosis, and inhibit AR translocation from the cytoplasm to the nucleus [11]. Combining treatments with different mechanisms of action often achieves the highest tumor regression [12].

A large body of evidence supports upfront combination treatment of ADT with an ARPI and/or docetaxel in men with mCSPC. However, real-world data show a significantly low rate of adoption of this combination regimen in clinical practice. This strategy is currently used in less than half of patients eligible for treatment intensification in many countries [13].

This review aims to present the scientific rationale and the most recent evidence for mCSPC treatment strategies, focusing on treatment intensification using ADT combined with ARPI and/or docetaxel. We aim to provide guidance on applying this evidence to ensure the appropriate use of all available treatment options in clinical practice. Furthermore, we discuss the contemporary treatment landscape for mCSPC, analyze the clinical evidence supporting the combined treatment approach, and identify key issues requiring further investigation. Thus, we seek to offer insights that will optimize treatment strategies and improve patient outcomes in mCSPC.

## 2. Theoretical Background

### 2.1. Targeting the Androgen Signaling Axis 

Androgens play a pivotal role in maintaining male physiology and prostate function. AR activation is central to the pathogenesis of prostate cancer, driving tumor growth and progression [14]. The seminal discovery by Charles Huggins and Clarence Hodges in the 1940s, demonstrating tumor regression after androgen deprivation, laid the foundation for targeting AR in prostate cancer therapy [15]. However, despite initial success, resistance to AR-directed therapies remains a significant challenge, particularly in metastatic prostate cancer, which remains largely incurable [16].

The AR is a ligand-dependent transcription factor belonging to the steroid receptor family. Under normal physiological conditions, testosterone and dihydrotestosterone (DHT) are its principal ligands. Ligand binding induces conformational changes in the AR, leading to its nuclear translocation and activation of target gene transcription, representing essential processes for prostate cancer progression [17].

ADT, achieved via surgical or medical castration, has been a cornerstone in prostate cancer treatment [18]. The development of next-generation ARPIs, such as abiraterone and enzalutamide, which provided survival benefits in CRPC, marked significant milestones [19]. Despite these advancements, tumor resistance has evolved, necessitating further exploration [20].

Prostate cancer can synthesize androgens intratumorally, maintaining AR signaling despite castrate serum testosterone levels [21]. Enzymes involved in androgen biosynthesis, such as CYP17A1 and AKR1C3, are upregulated in CRPC. Inhibitors targeting these pathways, such as abiraterone and other ARPIs, aim to disrupt androgen biosynthesis but face limitations due to the complexity and redundancy of steroidogenic pathways [22].

Targeting the downstream effects of AR activation, such as specific gene transcription and oncogenic pathways, offers another therapeutic avenue. AR signaling intersects with other pathways, such as PI3K/AKT and DNA damage repair mechanisms; thus, combination therapies might be able to overcome tumor resistance [23]. 

Figure 1 illustrates the mechanisms of androgen signaling and the therapeutic targets in the systemic treatment of advanced PC.

**Figure 1 cancers-16-03187-f001:**
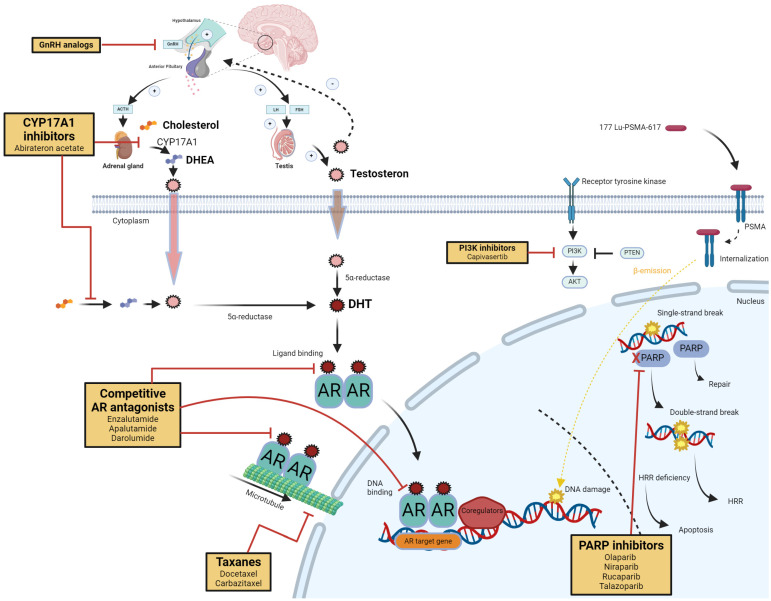
Mechanisms of androgen signaling and therapeutic targets in systemic treatment of advanced prostate cancer. Androgen signaling and the therapeutic targets of systemic therapies for advanced prostate cancer encompass a range of receptor-mediated and pre/post-receptor mechanisms, each serving as a focal point for distinct therapeutic strategies. The synthesis of androgens is precisely controlled by the hypothalamic–pituitary–gonadal and hypothalamic–pituitary–adrenal axes, which regulate the production of gonadal and adrenal androgens, precursors to DHT, the main ligand for the androgen receptor (pre-receptor activity). Upon ligand binding, the androgen receptor moves from the cytoplasm to the nucleus, where it binds to DNA as a homodimer, facilitating the transactivation of target genes and signaling pathways (post-receptor activity). The figure also highlights various clinically approved and experimental inhibitors. Abbreviations: ACTH, adrenocorticotropic hormone; AKT, AKR thymoma; AR, androgen receptor; ARE, androgen response element; CYP17A1, cytochrome P450 17A1; DHEA, dehydroepiandrosterone; DHT, dihydrotestosterone; DNA, deoxyribonucleic acid; FSH, follicle-stimulating hormone; GnRH, gonadotropin-releasing hormone; HRR, homologous recombinational repair; LH, luteinizing hormone; PARP, poly (ADP-ribose) polymerase; PI3K, phosphoinositide 3-kinase; PSA, prostate-specific antigen; PSMA, prostate-specific membrane antigen; PTEN, phosphatase and tensin homolog. Created with BioRender.com.

### 2.2. Combination Therapies

Combination therapy leverages multiple treatment modalities to enhance therapeutic efficacy and counteract resistance, which is particularly crucial in prostate cancer due to its heterogeneity and capacity to develop resistance [24]. By integrating surgery, radiation, hormone therapy, chemotherapy, and targeted therapies, combination therapy targets prostate cancer using multiple strategies, thereby increasing the likelihood of successful treatment outcomes [25]. The slow-growing nature of prostate cancer and its dependence on various growth factors make it particularly suitable for such a multimodal approach [26].

The heterogeneity within prostate tumors means that different tumor areas may respond variably to treatment [27]. Therefore, combination therapies can more effectively target all cancerous regions and reduce the risk of resistance [28]. Strategic combinations in cancer therapy address tumor complexity and heterogeneity by developing regimens targeting independent molecular pathways, thus overcoming resistance and enhancing patient outcomes [29]. Additionally, exploring epigenetic regulators and alternative androgen biosynthesis pathways offers new treatment opportunities [30].

The therapeutic landscape for prostate cancer continues to evolve with deeper insights into AR signaling and resistance mechanisms. While next-generation AR-directed therapies have improved outcomes, resistance remains a significant challenge [31,32]. Combination therapy presents a promising approach to enhancing treatment efficacy and improving patient outcomes by addressing the complexity and heterogeneity of prostate cancer [33,34]. Ongoing research on the molecular underpinnings of AR function and resistance is crucial for developing more effective treatments, ultimately aiming to transform prostate cancer into a manageable chronic condition [35,36].

## 3. Clinical Development for Combination Therapy

### 3.1. Systemic Treatment in mCSPC

#### 3.1.1. Androgen-Deprivation Therapy

AR inhibition remains the mainstay treatment for metastatic prostate cancer, a practice grounded in seminal experiments from 1941, which demonstrated the androgen-driven and androgen-dependent nature of prostate cancer [37]. Androgen signaling is crucial for prostate cancer growth and survival, even in treatment-resistant cases [36].

Testosterone suppression (TS) therapy initially involved surgical castration (bilateral orchiectomy) and diethylstilbestrol, eventually evolving to include luteinizing hormone-releasing hormone (LHRH) agonists and antagonists based on the understanding of hypothalamic–pituitary control of gonadal testosterone production [38]. Combining antiandrogens with ADT, known as complete androgen blockade, would counteract both testicular and adrenal androgens. Early-generation AR inhibitors, such as flutamide, bicalutamide, nilutamide, and cyproterone acetate, are typically not used alone but are combined with TS to prevent initial flare responses from LHRH agonists [39].

A meta-analysis of 8275 men from 27 trials comparing TS alone to combined ADT showed improved 5-year OS with nonsteroidal antiandrogens (absolute benefit 3%; two-sided *p* = 0.005) and potentially worse outcomes with cyproterone acetate (absolute reduction 3%; two-sided *p* = 0.04). These findings support the use of ADT combination with early-generation AR inhibitors as a control in mCSPC clinical trials, although real-world practices vary [7].

LHRH agonists or antagonists reduce serum testosterone to castration levels (≤50 ng/dL or ≤1.74 nmol/L) [40]. However, phase 3 trials in mCSPC revealed that patients on ADT alone often develop castration resistance within a year, with a median OS of 36–54 months [41]. Therefore, guidelines recommend ADT monotherapy only when combination therapy is not feasible [42]. Furthermore, ADT with treatment intensification is advised for mCSPC unless contraindications exist [13]. 

Adverse events (AEs) related to ADT use include hot flushes, gynecomastia, decreased libido, sexual dysfunction, bone loss, increased fracture risk, sarcopenic obesity linked to metabolic syndrome, diabetes, cardiovascular disease, cognitive and mood disturbances, anemia, and reduced testis and penis volume, as well as testosterone flares with LHRH agonists. Short-term nonsteroidal AR inhibitors with LHRH agonists to control testosterone flare, patient education on heart-healthy lifestyle changes, optimizing control of cardiovascular risk factors, and regular bone health assessments and interventions such as physical activity, calcium, vitamin D, and bone-supportive therapy are used to manage these AEs [43].

#### 3.1.2. Treatment Intensification—Doublet Therapies 

##### Abiraterone Acetate

Abiraterone acetate was evaluated in several clinical trials for mCSPC to determine its effectiveness in improving OS and other clinical outcomes when combined with ADT. In the STAMPEDE trial, around 1000 mCSPC patients were randomized to receive either ADT alone or ADT combined with abiraterone. After a 40-month follow-up, the combination of abiraterone + ADT reduced the risk of death by approximately 40% (hazard ratio (HR): 0.61; 95% confidence interval (CI): 0.49–0.75), with significant improvements in other secondary endpoints [44]. A post hoc analysis of 1003 metastatic patients, stratified by LATITUDE criteria, showed that median survival was higher in the abiraterone +ADT arm (79 months vs. 46 months) after a median follow-up of 73 months. Both low-risk (HR: 0.54; 95% CI: 0.40–0.74) and high-risk (HR: 0.54; 95% CI: 0.43–0.69) groups benefited from the treatment [45].

The LATITUDE trial included 1199 patients with high-risk mCSPC, defined as having at least two of the following: Gleason score ≥ 8, ≥3 bone lesions, or visceral metastases. Patients were randomized to receive abiraterone with prednisone or placebo with ADT. The trial demonstrated a similar approximately 40% reduction in the risk of death with abiraterone + ADT (HR: 0.66; 95% CI: 0.56–0.78; *p* < 0.001) after a median follow-up of 51.8 months. The median OS was 53.3 months in the abiraterone + ADT group compared to 36.5 months in the placebo group. Secondary endpoints, such as radiographic progression-free survival (rPFS), PSA progression, and time to chemotherapy initiation, also showed significant improvements [46].

The STAMPEDE trial demonstrated a high incidence of severe and life-threatening grade 3–5 AEs, particularly in combination therapy arms. Adding abiraterone significantly increased the frequency of these AEs. This finding underscores the necessity for a careful risk-benefit analysis when incorporating abiraterone into treatment plans for mCSPC, especially in patients aged ≥ 70 years who experienced higher incidences of grade 3–5 AEs (47% vs. 33%) and a higher rate of treatment-related deaths (9 vs. 3). Additionally, severe hypertension or cardiac disorders were observed in 10% of patients, and grade 3–5 liver toxicity was noted in 7% of patients, highlighting the importance of vigilant monitoring of blood pressure, as well as renal and hepatic function, to improve patient outcomes when using abiraterone in combination therapy [44].

These findings align with those reported in the LATITUDE trial, also demonstrating a higher frequency of AEs in the abiraterone group. However, these events were generally mild and primarily related to mineralocorticoid excess (hypertension, hypokalemia, and edema), hormonal effects (fatigue and hot flushes), and liver toxicity [44]. Despite these AEs, the overall discontinuation rate due to AEs in the LATITUDE trial was 12%. Importantly, patient-reported outcomes in LATITUDE indicated improvements in pain intensity, fatigue, functional decline, and overall health-related QoL, further emphasizing the complex balance between the therapeutic benefits and the risks of AEs when using abiraterone in mCSPC [47]. 

In February 2018, the FDA approved abiraterone in combination with prednisone for mCSPC based on the results of the STAMPEDE and LATITUDE trials. The National Comprehensive Cancer Network (NCCN) Panel recommends abiraterone with 5 mg once-daily prednisone as a treatment option with ADT for newly diagnosed mCSPC. Alternatively, the fine-particle formulation of abiraterone can be used to reduce financial toxicity and improve adherence [48].

##### Apalutamide 

The double-blind phase 3 TITAN trial evaluated the efficacy and safety of apalutamide in combination with ADT compared to ADT alone in patients with mCSPC [49]. This trial was designed to address the ongoing need for more effective treatment strategies in such patients. The TITAN trial randomized 1052 patients with mCSPC to receive either ADT with apalutamide (240 mg/day) or a placebo. The participants were carefully stratified based on Gleason score at diagnosis, geographic region, and previous docetaxel treatment to ensure balanced comparison groups. The median follow-up duration was 22.7 months, allowing for a robust assessment of long-term outcomes. Importantly, both primary endpoints were achieved, demonstrating the significant benefit of apalutamide in this setting. At 24 months, the rPFS was notably higher in the apalutamide group (68.2%) compared to the placebo group (47.5%), with an HR for radiographic progression or death of 0.48 (95% CI: 0.39–0.60; *p* < 0.001). Furthermore, OS was also significantly improved in the apalutamide group (82.4%) compared to the placebo group (73.5%), with an HR for death of 0.67 (95% CI: 0.51–0.89; *p* = 0.005). These findings underscore the potential of adding apalutamide to standard ADT to enhance outcomes in patients with mCSPC.

In the final analysis, the median OS was not reached in the apalutamide group compared to 52.2 months in the ADT alone group (HR: 0.65; 95% CI: 0.53–0.79; *p* < 0.001) after a median follow-up of 44 months. After adjusting for crossover, the risk of death was reduced by 48%, indicating a consistent benefit across both HV and low-volume (LV) disease. A post hoc subgroup analysis of the TITAN trial re-evaluated outcomes based on the volume and timing of metastases, showing synchronous HV disease in 54% of patients. OS benefits were statistically significant in patients with synchronous HV disease (HR: 0.68; 95% CI: 0.53–0.87; *p* = 0.002) and metachronous LV disease (HR: 0.22; 95% CI: 0.09–0.55; *p* = 0.001). However, no significant OS benefits were observed in patients with synchronous LV or metachronous HV disease, likely due to limited patient numbers [50].

More frequently observed AEs in the apalutamide group included rash, hypothyroidism, and ischemic heart disease. Nevertheless, health-related QoL was maintained during treatment with apalutamide [51]. 

The TITAN trial demonstrated significant improvements in both rPFS and OS by adding apalutamide to ADT in patients with mCSPC. These findings support apalutamide as a category 1 treatment option for mCSPC, which received FDA approval in September 2019 [52]. The consistent survival benefits across different subgroups and the maintained QoL highlight the robust efficacy and tolerability of apalutamide in this patient population [53].

##### Enzalutamide 

The management of mCSPC has evolved with the introduction of enzalutamide, an ARPI, as an addition to standard ADT. Recent clinical trials, namely ENZAMET and ARCHES, have provided significant evidence of the efficacy and safety of this combination [8].

The ENZAMET trial was an open-label, randomized phase 3 study involving 1125 patients with mCSPC. The patients were randomized to receive either enzalutamide (160 mg/day) + ADT (LHRH analog or surgical castration) or a first-generation antiandrogen (bicalutamide, nilutamide, or flutamide) + ADT [54]. Stratification was based on disease volume, planned use of early docetaxel, planned use of bone antiresorptive therapy, comorbidity score, and trial site. At the first interim analysis with a median follow-up of 34 months, the enzalutamide group showed a 33% reduction in the risk of death compared to the control group (HR: 0.67; 95% CI: 0.52–0.86; *p* = 0.002). An analysis after a median follow-up of 68 months revealed that the 5-year OS rate was 67% in the enzalutamide group versus 57% in the control group (HR: 0.70; 95% CI: 0.58–0.84; *p* < 0.001). Secondary endpoints also showed improvement with enzalutamide, including PFS based on PSA levels and clinical PFS. A post hoc analysis indicated a significant OS benefit in patients with synchronous mCSPC (HR: 0.73; 95% CI: 0.55–0.99), but not in metachronous patients [55].

The ARCHES trial, a double-blind, randomized phase 3 study, included 1150 patients with mCSPC. The participants were randomized to receive either enzalutamide (160 mg/day) + ADT or placebo + ADT, with stratification based on disease volume and prior docetaxel use. After a median follow-up of 14.4 months, the enzalutamide group demonstrated significantly improved rPFS (19.0 months vs. not reached; HR: 0.39; 95% CI: 0.30–0.50; *p* < 0.001) [56]. At the final OS analysis, although 32% of patients in the placebo group crossed over to enzalutamide after unblinding, enzalutamide still reduced the risk of death by 34% compared to placebo (HR: 0.66; 95% CI: 0.53–0.81; *p* < 0.001) [57].

The safety of enzalutamide in these trials was consistent with previous studies on CRPC, with common AEs including fatigue, seizures, and hypertension. The safety profiles observed in both ENZAMET and ARCHES trials were similar, confirming the manageable safety of enzalutamide in mCSPC [54,56].

Overall, enzalutamide, when combined with ADT, significantly improves survival in patients with mCSPC. The ENZAMET and ARCHES trials demonstrated substantial improvements in OS and PFS with enzalutamide + ADT compared to standard ADT regimens. The FDA’s approval of enzalutamide for mCSPC in December 2019 reflects its robust efficacy and manageable safety profile, supporting its use as a category 1 treatment option for patients with mCSPC [58].

##### Docetaxel

Docetaxel has been studied as an initial treatment option for patients with mCSPC based on several key phase 3 trials: ECOG 3805/CHAARTED, STAMPEDE, and GETUG-AFU-15 [41,59,60]. These studies provided significant insights into the efficacy of combining docetaxel with ADT for improving OS in patients with mCSPC.

The ECOG 3805/CHAARTED trial randomized 790 patients with mCSPC to receive either docetaxel (75 mg/m^2^ intravenously every 3 weeks for 6 doses) + ADT or ADT alone [41]. After a median follow-up of 53.7 months, the combination therapy arm showed longer OS than the ADT-alone arm (57.6 months vs. 47.2 months; HR: 0.72; 95% CI: 0.59–0.89; *p* = 0.002) [61]. Subgroup analysis revealed that the survival benefit was more pronounced in patients with HV disease (65% of participants) (HR: 0.63; 95% CI: 0.50–0.79; *p* < 0.001). However, patients with LV did not obtain a significant survival benefit by adding docetaxel (HR: 1.04; 95% CI: 0.70–1.55; *p* = 0.860).

The STAMPEDE trial, a multi-arm, multi-stage phase 3 study, included patients with both non-metastatic (M0) and metastatic (M1) CSPC [59]. The trial confirmed the survival advantage of adding docetaxel to ADT, as seen in the CHAARTED trial. In the STAMPEDE trial, the extent of disease was not evaluated in 1087 patients with metastatic disease, but the median OS for all M1 patients was 5.4 years in the ADT + docetaxel arm versus 3.6 years in the ADT-only arm (a difference of 1.8 years compared to a 1.1-year difference in the CHAARTED).

The GETUG-AFU-15 trial evaluated ADT combined with docetaxel [62]. After 83.9 months of follow-up, this combination did not significantly improve OS compared to ADT alone. However, post hoc analysis considering metastatic disease volume indicated a trend toward benefit in the HV group (HR: 0.78; 95% CI: 0.56–1.09; *p* = 0.140) [60].

A meta-analysis of three trials involving 2,261 patients with mHSPC revealed that the combination of ADT with docetaxel did not improve OS in men with metachronous LV disease, as indicated by an HR of 0.98 (95% CI: 0.67–1.45). However, the most significant OS benefit and reduction in the risk of death were observed in patients with synchronous HV disease (HR: 0.60; 95% CI: 0.52–0.69). Thus, while ADT combined with docetaxel markedly improves OS in patients with HV mCSPC, particularly in those with synchronous HV disease, a different systemic therapy approach may be necessary for patients with metachronous LV disease [63].

Additionally, triplet therapy options including ADT, docetaxel, and ARPI have improved OS compared to ADT + docetaxel alone [64]. Consequently, ADT and docetaxel are not recommended for patients with LV mCSPC, whereas triplet therapy is advised for medically fit patients with HV disease.

The key clinical trials discussed in this section, along with their outcomes and adverse events, are summarized in Table 1 for a comprehensive comparison.

#### 3.1.3. Treatment Intensification—Triplet Therapies

##### Rationale

Numerous trials demonstrated the significant OS benefit of beginning systemic therapy earlier at the mCSPC stage, not waiting until the tumor becomes castration-resistant [8,65,66]. This benefit is likely due to several reasons: mCSPC might have more favorable disease biology, less acquired treatment resistance, and therefore demonstrates a more durable treatment response [8]. Using docetaxel in mCSPC is likely more efficacious at targeting AR-independent cancer cells early compared to its use in mCRPC since these cancer cells might have had opportunities to develop resistance [8,67]. Treatment in the mCSPC setting is also often better tolerated given fewer cumulative toxicities from prior systemic therapy, as well as fewer symptoms and a lower disease burden before disease progression. Real-world studies showed that only approximately 50% of patients receive second-line therapies upon disease progression, with many possibly becoming too frail to receive docetaxel in the mCRPC setting. Therefore, earlier treatment intensification may also increase the number of patients receiving life-prolonging systemic therapy [68].

The effectiveness of chemohormonal triplet therapy may stem from the synergistic and complementary effects of the three agents [69]. ADT, the cornerstone of hormonal therapy for prostate cancer, inhibits tumor growth by lowering androgen levels produced in the testicles through medication or surgery. ARPIs, including androgen synthesis inhibitors (abiraterone) and antiandrogens (enzalutamide, apalutamide, and darolutamide), suppress other androgen sources to achieve the maximum suppression of the androgen axis, further improving the anti-tumor effect [28]. Docetaxel, a semisynthetic taxane, exhibits significant antitumor activity by inhibiting microtubular depolymerization and attenuating the effects of BCL-2 and BCL-XL expression. The taxane-induced microtubule stabilization arrests cells in the G2/M phase and induces bcl-2 phosphorylation, promoting a cascade of events that ultimately cause apoptotic cell death [70]. 

ADT lowers androgen levels, inhibiting the growth of androgen-sensitive CSPC cells that respond well to this treatment [9]. The combination of docetaxel and ADT can create a synergistic effect by simultaneously inhibiting androgen signaling pathways and cell division [71]. Clinical trials, such as CHAARTED and STAMPEDE, showed that combining docetaxel with ADT significantly improves survival in patients with mCSPC, providing clinical evidence of the effectiveness of docetaxel against mCSPC [41,72]. 

##### Docetaxel Plus Abiraterone

The PEACE-1 trial was an international, open-label, randomized phase 3 study conducted across seven European countries, aiming to assess the efficacy of adding abiraterone acetate with prednisone to the standard of care (SOC) in patients with de novo mCSPC [73]. The study utilized a 2 × 2 factorial design and included 1173 patients randomized into four groups: SOC ADT alone or with docetaxel, SOC with radiotherapy (RT), SOC with abiraterone, and SOC with RT and abiraterone. The primary endpoints were rPFS and OS. The results showed that patients receiving abiraterone had significantly longer rPFS compared to those who did not. Specifically, the overall rPFS in the abiraterone group had an HR of 0.54 (99.9% CI, 0.41–0.71; *p* < 0.001). For those receiving abiraterone with docetaxel, the rPFS had an HR of 0.50 (99.9% CI, 0.34–0.71; *p* < 0.001). Furthermore, adding abiraterone to SOC significantly improved OS, with the overall OS in the abiraterone group showing an HR of 0.82 (95.1% CI, 0.69–0.98; *p* = 0.030). In patients receiving abiraterone with docetaxel, the OS had an HR of 0.75 (95.1% CI, 0.59–0.95; *p* = 0.017). 

Subgroup analysis provides significant insights into the efficacy of different treatment combinations for mCSPC based on disease burden. For patients with HV disease, adding abiraterone acetate + prednisone to the SOC (consisting of ADT and docetaxel) markedly improved OS and rPFS. Specifically, the median OS for patients receiving the triplet therapy was 5.1 years compared to 3.5 years for those receiving only ADT plus docetaxel, with an HR of 0.72 (95% CI: 0.55–0.95; *p* = 0.019). Additionally, the median rPFS was 4.1 years for those receiving the triplet therapy versus 1.6 years for those receiving the SOC alone (HR: 0.47; 95% CI: 0.30–0.72; *p* < 0.001). In contrast, the OS data remain immature for patients with LV disease, and median OS has not yet been reached in either treatment group. The rPFS also showed significant improvement with the triplet therapy: the median rPFS was not reached versus 2.7 years for the SOC alone (HR: 0.58; 95% CI: 0.29–1.15; *p* = 0.006).

The study also examined treatment-related AEs, finding that rates of neutropenia, febrile neutropenia, fatigue, and neuropathy were similar between triplet and doublet therapy groups. However, the incidence of grade ≥ 3 AEs was higher in the triplet therapy group (63% vs. 52%). Specifically, grade 3 hypertension and grade 3 transaminase increase were more common in the triplet group (22% vs. 13% and 6% vs. 1%, respectively). Other AEs, such as febrile neutropenia, fatigue, and neuropathy, did not show significant differences between the groups.

The PEACE-1 trial demonstrated that adding abiraterone to ADT and docetaxel significantly improves both rPFS and OS in patients with de novo mCSPC. These results emphasize the substantial benefit of triplet therapy for HV mCSPC. Thus, it should be considered a standard treatment option for this subgroup. For LV disease, while the survival benefit is still being evaluated, triplet therapy shows promise, particularly in improving PFS. 

##### Docetaxel Plus Darolutamide

The ARASENS trial, an international phase 3 study, evaluated the efficacy and safety of adding darolutamide to ADT and docetaxel in patients with mCSPC. This trial randomized 1306 patients to receive either the combination of ADT, docetaxel, and darolutamide or ADT, docetaxel, and a matching placebo [74]. The primary endpoint of the study was OS, with time to CRPC, skeletal event-free survival, and time to initiation of subsequent systemic antineoplastic therapy as secondary endpoints. The study did not stratify patients by disease volume, and 86% of participants had synchronous metastatic disease. The results demonstrated that adding darolutamide significantly improved OS compared to the placebo group. At 4 years, the OS rate was 62.7% (95% CI: 58.7–66.7) in the darolutamide group versus 50.4% (95% CI: 46.3–54.6) in the placebo group, reflecting a 32% reduction in the risk of death (HR: 0.68; 95% CI: 0.57–0.80; *p* < 0.001). Additionally, darolutamide showed significant benefits across several secondary endpoints. The time to CRPC was markedly prolonged (HR: 0.36; 95% CI: 0.30–0.42; *p* < 0.001), skeletal event-free survival improved (HR: 0.61; 95% CI: 0.52–0.72; *p* < 0.001), and the time to initiation of subsequent systemic antineoplastic therapy was extended (HR: 0.39; 95% CI: 0.33–0.46; *p* < 0.001).

Post hoc analyses demonstrated the OS benefit for patients with HV disease (HR: 0.69; 95% CI: 0.57–0.82), while the benefit was less clear in patients with LV disease (HR: 0.68; 95% CI: 0.41–1.13), as defined by the CHAARTED criteria. Additionally, the trial provided clear evidence of survival benefits for both high-risk patients (HR: 0.71; 95% CI: 0.58–0.86) and low-risk patients (HR: 0.62; 95% CI: 0.42–0.90) according to the LATITUDE criteria.

AEs were comparable between darolutamide and placebo groups, with most AEs being known effects of docetaxel. The most common AEs included alopecia, neutropenia, fatigue, and anemia. Darolutamide was associated with a higher incidence of rash (16.6% vs. 13.5%) and hypertension (13.7% vs. 9.2%), consistent with the known effects of ARPIs.

The findings of the ARASENS trial support the addition of darolutamide to ADT and docetaxel for patients with mCSPC, demonstrating significant improvements in OS and secondary endpoints. The consistency of the OS benefit across most subgroups underscores the robustness of these findings. However, the less pronounced benefit in patients with LV disease warrants further research with larger sample sizes and longer follow-ups. The ARASENS trial established darolutamide in combination with ADT and docetaxel as a superior treatment regimen for mCSPC, offering a substantial survival advantage and delayed disease progression. Based on these results, the FDA approved this triplet therapy in August 2022, marking a significant advancement in mCSPC management [75].

##### Docetaxel Plus Enzalutamide

The ENZAMET (Enzalutamide for Metastatic Prostate Cancer) study investigated the concurrent use of docetaxel with enzalutamide, with 45% of patients receiving planned docetaxel at the investigator’s discretion. Additionally, 85% of patients in the control arm received subsequent therapy, including 76% who received either abiraterone or enzalutamide upon progression [55]. A prespecified analysis demonstrated a significant difference in OS, favoring the enzalutamide arm among the subset of 362 men with synchronous metastatic disease planned for docetaxel, with a 5-year OS of 60% compared to 52% (HR: 0.73; 95% CI: 0.55–0.99). This survival benefit was not observed in patients with metachronous disease planned for docetaxel (HR: 1.10; 95% CI: 0.65–1.86). Within the synchronous population planned for docetaxel, OS estimates favored enzalutamide in both HV and LV subgroups. Survival curves indicated higher OS rates in the first 30 months for participants receiving enzalutamide + docetaxel + testosterone suppression (TS) versus those contemporaneously accrued to enzalutamide + TS in the highest-risk subgroup (synchronous, HV), highlighting the potential necessity of early chemotherapy in rapidly lethal disease. The AEs of adding enzalutamide to the SOC were overall similar to those of enzalutamide in previous clinical trials.

These findings provide robust evidence supporting the use of enzalutamide in combination with ADT and docetaxel, particularly for patients with synchronous metastatic disease, indicating a significant improvement in OS and underscoring the importance of early intervention in high-risk populations.

Table 2 provides a summary of key clinical trials evaluating triplet therapies, highlighting their efficacy, patient populations, and key findings.

#### 3.1.4. Network Meta-Analysis

Until recently, several network meta-analyses (NMA) have been conducted to update the existing evidence on the comparative efficacy of systemic therapy in mCSPC by prognostic subgroups, aiming to support clinical practice guidelines. Among these studies, one conducted by Hoeh et al. focused on evaluating the efficacy of triplet versus doublet therapies in mCSPC, specifically stratified by disease volume (low vs. high) [76]. Their analysis, which incorporated data from 10 randomized controlled trials (RCTs), centered on OS outcomes for various treatment regimens, including doublet therapy (ARPI + ADT or docetaxel + ADT) and triplet therapy (ARPI + docetaxel + ADT). The NMA was conducted separately for patients with LV and HV mCSPC based on the CHAARTED criteria. Their findings revealed that combination therapies other than ARPI + ADT did not show substantial benefits compared to ADT alone for LV mCSPC. Moreover, no significant OS differences were observed between triplet therapies and the ARAPI + ADT doublet therapy. In contrast, all combination therapies improved OS compared to ADT alone for HV mCSPC. Notably, the triplet regimen of darolutamide + docetaxel + ADT ranked the highest regarding the OS benefit (*p* = 0.920), followed closely by abiraterone + docetaxel + ADT (*p* = 0.850). Specifically, darolutamide + docetaxel + ADT showed a significant advantage in terms of OS over ARAT + ADT (HR: 0.76; 95% CI: 0.59–0.97). This study underscores the necessity of stratifying patients by disease volume when making treatment decisions for mCSPC. Therefore, while triplet therapy may not provide significant OS benefits for patients with LV mCSPC, it offers superior outcomes for those with HV disease, thereby emphasizing the importance of personalized treatment strategies based on disease burden.

Building on this, Jian et al. conducted a systematic review and NMA encompassing 18 publications from 12 clinical trials to further compare the efficacy of currently available combination therapies in patients with mCSPC [69]. The overall findings corroborated those of Hoeh et al., with triplet therapy ranking first in terms of OS (HR: 0.57; 95% credible interval (CrI): 0.48–0.67) and rPFS (HR: 0.33; 95% CrI: 0.26–0.41) benefits in the general mCSPC population. For HV mCSPC, triplet therapy was also ranked first in OS (HR: 0.57; 95% CrI: 0.44–0.75) and rPFS (HR: 0.29; 95% CrI: 0.23–0.37) benefits, followed by the doublet therapy of ADT + rezvilutamide (OS: HR: 0.58; 95% CrI: 0.44–0.77; rPFS: HR: 0.44; 95% CrI: 0.33–0.58) and ADT + docetaxel (OS: HR: 0.75; 95% CrI: 0.62–0.91; rPFS: HR: 0.63; 95% CrI: 0.52–0.77). In LV mCSPC, the combination of ADT with ARAT ranked first in OS (HR: 0.68; 95% CrI: 0.58–0.80) and rPFS (HR: 0.50; 95% CrI: 0.42–0.60) benefits, with ADT + apalutamide being the top therapy for OS (HR: 0.53; 95% CrI: 0.35–0.79) and ADT + enzalutamide showing significant improvement in OS and rPFS (OS: HR: 0.56; 95% CrI: 0.40–0.77; rPFS: HR: 0.29; 95% CrI: 0.22–0.39). However, triplet therapies did not demonstrate improvements in OS or rPFS in LV disease (OS: HR: 0.81; 95% CrI: 0.60–1.08; rPFS: HR: 0.67; 95% CrI: 0.50–0.91) and were associated with a higher risk of AEs (any AE: odds ratio (OR): 2.50; 95% CrI: 1.80–3.50) and grade ≥ 3 AEs (OR: 3.20; 95% CrI: 2.40–4.30) compared to other therapies, further suggesting the need for careful consideration of disease volume when selecting treatment modalities.

Complementing these findings, Dr. Riaz et al. comprehensively evaluated contemporary systemic treatment options for patients with mCSPC [77]. This study synthesized data from ten phase 3 RCTs involving 11,043 patients, assessing key outcomes, such as OS, PFS, grade ≥ 3 AEs, and health-related QoL. The results reinforced the benefits of triplet therapies, such as darolutamide combined with docetaxel and ADT and abiraterone with prednisone combined with docetaxel and ADT, in improving OS compared to docetaxel combined with ADT but did not show significant benefits over ARPI doublets, such as abiraterone with prednisone combined with ADT, enzalutamide combined with ADT, and apalutamide combined with ADT. Specifically, the HR for the darolutamide triplet and the abiraterone with prednisone triplet was 0.68 (95% CI: 0.57–0.81) and 0.75 (95% CI: 0.59–0.95), respectively, compared to docetaxel combined with ADT. Subgroup analyses further highlighted that abiraterone with prednisone combined with docetaxel and ADT provided an OS advantage for patients with HV disease compared to docetaxel combined with ADT (HR: 0.72; 95% CI: 0.55–0.95) but not ARPI doublets. Conversely, triplet therapies did not significantly outperform ARPI doublets or docetaxel combined with ADT for LV disease. Moreover, the increased risk of grade ≥ 3 AEs associated with triplet therapies underscores the importance of balancing efficacy with safety when considering treatment options. Overall, these studies collectively emphasize the critical role of disease volume stratification in guiding treatment decisions for mCSPC.

Table 3 summarizes the key findings from network meta-analyses evaluating the efficacy of combination therapies in mCSPC, stratified by disease volume.

### 3.2. Radiotherapy in mCSPC

The treatment for the primary tumor in metastatic disease is an evolving strategy aimed at eliminating significant sources of lethal metastatic seeding [79,80]. Multiple clinical trials have investigated the efficacy of prostate RT in patients with mCSPC, particularly focusing on outcomes related to failure-free survival (FFS), OS, and QoL.

The STAMPEDE trial randomly assigned 2061 men to receive SOC (ADT with concurrent docetaxel permitted from late 2015) or SOC + prostate RT [25]. The primary outcomes demonstrated that prostate RT significantly improved FFS but did not improve OS in the overall cohort (HR: 0.92; 95% CI: 0.80–1.06; *p* = 0.226). However, a pre-planned analysis revealed a pronounced OS benefit in patients with a low metastatic burden (HR: 0.68; 95% CI: 0.52–0.90; *p* = 0.007), which was not evident in high-burden disease (interaction *p* = 0.010). This benefit was consistent in long-term follow-up, with no evidence of deterioration in global QoL or long-term high-grade urinary toxicity [81].

The HORRAD trial randomized de novo mCSPC patients to either ADT alone or ADT + RT using two RT schedules: 70 Gy in 35 fractions over 7 weeks or 57.76 Gy in 19 fractions over 6 weeks [82]. The trial showed a modest improvement in time to PSA progression favoring the RT arm (median, 15 vs. 12 months, HR: 0.78; 95% CI: 0.63–0.97; *p* = 0.020) but no OS benefit.

The PEACE-1 phase 3 trial assessed the survival benefit of adding prostate RT in men with low-burden disease receiving intensified systemic treatment (docetaxel and/or abiraterone). The trial found that combining prostate RT with intensified systemic treatment improved rPFS (HR: 0.50; 95% CI: 0.28–0.88; *p* < 0.001) and CRPC-free survival (HR: 0.32; 95% CI: 0.23–0.44; *p* < 0.001) in men with low-burden mCSPC, although no OS improvement was detected. Additionally, early prostate RT prevented severe urologic morbidity, irrespective of metastatic burden.

Meta-analyses combining data from trials such as HORRAD and STAMPEDE, as well as secondary analyses of PEACE-1, reinforced the potential benefits of prostate RT in patients with low-burden disease. Specifically, these analyses highlighted improved survival outcomes by adding RT to the SOC, particularly in patients with fewer bone metastases [83]. Ongoing trials, such as PEACE-6 Oligo (PRESTO) and PLATON, are evaluating the role of stereotactic radiotherapy (SBRT) in oligometastatic CSPC patients. These studies aim to further define the benefits of targeted RT to metastatic sites in combination with the SOC [84,85].

The evidence from these trials supports the proactive treatment of the primary tumor with RT in patients with low-burden mCSPC. The differential impact of metastatic burden on treatment outcomes underscores the need for personalized treatment strategies. While prostate RT improves FFS and reduces CRPC incidence in low-burden disease, its role in combination with systemic therapies, such as abiraterone and docetaxel, warrants further exploration. Prostate RT is an established standard for synchronous, low-burden/volume mCSPC, offering significant benefits in terms of FFS and QoL without compromising OS in the overall population [81]. 

## 4. Guidelines

Various professional organizations and associations have modified their guidelines for the optimal management of patients with mCSPC based on compelling evidence demonstrating the effectiveness of early combination therapy involving ADT with ARPIs or docetaxel [48,86]. Considering the robust evidence from the above-mentioned trials, the current NCCN and EAU recommendations advocate for (1) ADT in conjunction with abiraterone, apalutamide, or enzalutamide; (2) ADT in combination with docetaxel along with abiraterone or darolutamide; or (3) ADT with external beam RT to the primary tumor for low metastatic burden as the first-line treatment for mCSPC. 

## 5. Considerations

### 5.1. Real-World Evidence and Patterns

A recent Austrian multicenter study evaluated the effectiveness and tolerability of triplet therapy (ADT, ARPI, and chemotherapy) in treating mCSPC in a real-world setting [87]. This study involved 97 patients from 16 Austrian medical centers, with the majority receiving abiraterone (79.4%) and darolutamide (17.5%). Simultaneous administration of chemotherapy and ARPI significantly improved treatment outcomes, with a 99% decline in PSA levels in all patients and imaging responses in 88% of patients receiving abiraterone and 75% of patients receiving darolutamide. AEs were observed in 61.9% of patients, with 15% experiencing grade 3–5 events, including fatigue, dermatologic issues, infections, and polyneuropathy. Notably, starting ARPI before chemotherapy was associated with progression to more advanced disease. The study underscores the high effectiveness and tolerability of triplet therapy, especially for HV synchronous mCSPC, and highlights the importance of concurrent administration for achieving optimal outcomes.

Similarly, a recent study presented at the 2024 ASCO conference examined the outcomes of first-line triplet therapy (ADT, docetaxel, and abiraterone) in 58 Indian patients with mCSPC treated between May 2022 and December 2023 [88]. The patient cohort primarily comprised individuals with de novo metastatic disease (93.1%), an average age of 64 years, HV disease (94.8%), and a Gleason score ≥ 8 (69%). Comorbidities were prevalent in 89.7% of the patients, and 90% had an Eastern Cooperative Oncology Group (ECOG) performance status of 0–1. The median number of triplet therapy cycles administered was six. Grade ≤ 3 AEs were reported in 22.4% of patients, including hypertension (6.9%), febrile neutropenia (5.2%), hyperglycemia (5.2%), and urinary tract infection (3.4%), with no treatment-related deaths. Dose reductions were necessary in 24.1% of cases. Notably, 62% of patients achieved an undetectable PSA level (<0.2 ng/mL), with a 12-month OS rate of 98.3% and a biochemical PFS rate of 96.6%. Thus, triplet therapy is effective and well tolerated in Indian mCSPC patients, with low rates of serious AEs and high survival rates, supporting its real-world applicability.

Furthermore, another study presented at the 2024 ASCO conference aimed to assess the real-world application of guidelines recommending ARPIs and/or docetaxel with ADT for mCSPC in the US from 2017 to 2023 [89]. Using data from the Komodo Research Dataset (from January 2017 to September 2023), the retrospective cohort study included 10,717 men with a median age of 65 years. Findings indicated that 28% of patients received ARPIs, 9% of patients received docetaxel, and 2.5% of patients received both. From 2017 to 2023, ARPI use rose from 13% to 47%, and ARPI + docetaxel use increased from 0.8% to 15%, while the use of ADT + docetaxel and ADT alone declined. Factors such as younger age, de novo mCSPC, bone-only metastases, and opioid use were associated with higher treatment intensification. Despite increased adoption of ARPI and/or docetaxel, over a third of patients still received ADT alone, highlighting areas for improvement in clinical practice.

These three studies collectively underscore the effectiveness and tolerability of triplet therapy in treating mCSPC, particularly highlighting its critical success and the importance of concurrent administration in real-world clinical settings. Additionally, they show the increasing adoption of these therapies in clinical practice, while indicating the need for further improvement to ensure broader adherence to treatment guidelines. This body of evidence supports the critical role of triplet therapy in achieving optimal outcomes for mCSPC patients and underscores the potential for enhancing clinical practice based on these findings.

A systematic review explored the real-world application of clinical trial findings in the treatment of mCSPC, highlighting significant advances since 2015, including docetaxel chemotherapy and ARPI alongside ADT [90]. Analyzing 13 studies encompassing 166,876 patients, the review aimed to determine treatment utilization rates and identify influencing factors. The utilization rates for treatment intensification with docetaxel or neoadjuvant hormonal therapy (NHT) ranged from 9.3% to 38.1%. Younger, white, urban-dwelling patients and those treated in private or academic settings were more likely to receive these treatments, with oncologists more inclined to prescribe NHT than urologists. Temporal trends showed increasing NHT utilization but a decline in docetaxel use after 2015–2016, revealing a significant gap between clinical trial evidence and real-world practice. This underscores the need for improved strategies to facilitate the adoption of evidence-based treatments in routine clinical care.

### 5.2. Factors Influencing the Low Uptake of Combination Therapy

Despite high-level evidence and explicit guideline recommendations for combination therapy in mCSPC, the uptake in real-world settings remains significantly lacking [13,91,92]. Several factors contribute to this discrepancy, including cost and reimbursement models, availability and geographical variation, age and treatment setting, regulatory and reimbursement approvals, physician decision-making and practice patterns, and resource limitations in developing countries. The financial burden and variability in reimbursement models significantly influence patient access to therapy [13]. For instance, although being a cost-effective option, docetaxel use in the US remains low, likely due to cost-related issues [65,92]. Furthermore, the availability of therapeutic agents and health economics vary greatly by geographical location, affecting access to combination therapy [93]. While younger patients and those treated in private hospitals are more likely to receive combination therapy, overall uptake remains low across different age groups and settings. Delays and differences in the approval and reimbursement of new drugs across countries further contribute to varied uptake rates. Additionally, clinicians’ decisions, often influenced by local practice patterns and resource availability, impact the low adoption rates. Even in high-income countries, adherence to guidelines is inconsistent, indicating a gap between evidence-based recommendations and clinical practice [94,95]. Resource constraints significantly impact the ability to implement combination therapies in developing countries, dictating the preference for more affordable options such as ADT alone or ADT combined with docetaxel [96].

Real-world data underscore the complexities of implementing new treatment standards. The adoption of combination therapy for mCSPC follows a pattern described by the diffusion of innovation theory, with early adopters leading the way (Figure 2). However, widespread adoption is hindered by several factors. The adoption process starts with innovators and early adopters, followed by a gradual uptake by the majority. Currently, the uptake of combination therapy is in the early adoption phase, despite substantial evidence supporting its benefits. Although the efficacy of combination therapies has been well documented for over half a decade, broad adoption has not yet been observed. The uro-oncology community might require more time to fully integrate these treatments into standard practice. Clinicians must maintain high standards of care, ensuring that evidence-based practices are upheld and not compromised due to slower adoption rates. These considerations highlight the need for ongoing efforts to narrow the gap between clinical evidence and practice, ensuring that patients receive optimal, evidence-based care [13].

## 6. Ongoing Issues and Trials in mCSPC Treatment

### 6.1. What Is the Optimal Agent to Use Following Triplet Therapy

The treatment landscape for mCSPC is rapidly evolving, particularly in the context of post-triplet therapy options. The advent of novel agents, such as cabazitaxel, Lutetium-177 PSMA-617 (LuPSMA), and PARP inhibitors (PARPis), for patients with homologous recombination repair mutations (HRRm) underscores the necessity for identifying the most effective and safe post-triplet therapy agents [65].

Cabazitaxel, a next-generation taxane, demonstrated substantial efficacy in mCRPC post-docetaxel settings [97]. Its ability to overcome resistance mechanisms to other taxanes and its manageable toxicity profile make it a strong candidate for further exploration. Notably, its application to overcome taxane resistance is particularly relevant for patients who have progressed on prior therapies [98]. Targeted radioligand therapy with LuPSMA showed promise in mCRPC, especially in heavily pre-treated patients [99]. The TheraP trial highlighted its potential, demonstrating favorable responses in patients who have progressed after chemotherapy [100]. The capacity of this therapy to target PSMA-positive disease presents a significant advancement in personalized treatment [101]. For patients with HRRm, PARPis represent a targeted therapeutic approach exploiting the concept of synthetic lethality of cancer cells deficient in HRR. Ongoing trials, such as TALAPRO-3 and AMPLITUDE, are currently evaluating the efficacy of PARPis in the mCSPC setting. These agents hold promise for significantly impacting the treatment paradigm in HRRm patients [102].

To summarize, while there is no universal answer to the optimal agent following triplet therapy in mCSPC, the emerging evidence suggests that cabazitaxel, LuPSMA, and PARPis each offer distinct advantages depending on the patient’s specific disease characteristics and prior treatment history. The ability of cabazitaxel to overcome resistance, the targeted approach of LuPSMA to PSMA-positive disease, and the potential of PARPis in HRRm patients highlight the importance of personalized treatment. Further research and ongoing clinical trials will be crucial in refining these options and identifying the most effective post-triplet therapy agent for individual patients.

In line with the evolving treatment landscape for mCSPC, several ongoing clinical trials are exploring novel therapeutic combinations and approaches that build upon the current SOC. Noteworthy ongoing trials include CASCARA, which is investigating the efficacy of a quadruplet therapy regimen comprising ADT combined with cabazitaxel, carboplatin, and abiraterone in patients with HV mCSPC [103]. The CASCARA trial is designed to evaluate whether this intensified approach can provide superior outcomes in patients with more aggressive disease profiles. Another important trial, UpFront PSMA, focuses on the potential benefits of incorporating LuPSMA in mCSPC treatment [104]. By targeting PSMA-positive disease, LuPSMA may offer a more precise and effective treatment option in this setting. The UpFront PSMA trial will provide critical insights into the role of PSMA-targeted therapies early in the treatment course of mCSPC. Additionally, the AMPLITUDE trial assesses the efficacy of PARP inhibitors in mCSPC patients with HRRm [105]. Given the promising results of PARPis in other settings, the AMPLITUDE trial seeks to determine whether these agents can offer significant benefits when introduced earlier in treatment, particularly for patients harboring HRRm. 

### 6.2. Post-Triplet Therapy Strategy for Patients with HRRm

For patients with mCRPC and HRRm post-triplet therapy, the debate between PARPi monotherapy and the combination of PARPis with ARPIs is ongoing [106,107]. PARPi monotherapy leverages synthetic lethality in HRR-deficient cells, with initial studies indicating significant efficacy and manageable safety profiles. The combination of PARPis with ARPIs may provide synergistic effects, enhancing antitumor activity, although the risk of compounded toxicities necessitates careful patient selection and management [108]. Early data from trials, such as TALAPRO-3, suggest potential benefits [109].

### 6.3. Comparative Efficacy of Triplet Therapy vs. PARPis in the HRR-Mutated Population

The comparative efficacy of triplet therapy versus PARPis in the HRR-mutated population is an area of active research. Ongoing trials, such as TALAPRO-3 and AMPLITUDE, are expected to provide critical insights [66,109]. Preliminary data suggest that PARPis could offer a significant advantage in patients harboring HRRm, potentially altering the SOC in this subset. Research showed that suppressing HRR expression by androgen receptor inhibition increases cell vulnerability to PARPis, leading to PARP-dependent DNA damage-induced cell death [110].

### 6.4. The Need for Head-to-Head Trials

The combination of ADT with ARPIs versus the triplet combination of ADT, docetaxel, and ARPIs remains a critical area of investigation [111]. Currently, direct comparisons between these therapeutic strategies are limited. Hence, future research should focus on head-to-head trials to elucidate the benefits and drawbacks of these approaches.

### 6.5. Novel Emerging Therapeutic Agents 

Lutetium-177 PSMA-617 treatment yielded encouraging results in the VISION and TheraP trials, involving mCRPC patients previously treated with ARPIs and taxane regimens who had PSMA-positive disease [99,100]. Patients receiving radioligand therapy demonstrated improved rPFS and OS in the VISION trial, along with significant PSA response in the TheraP trial. Building on these results, the efficacy of targeted radionuclide therapy as part of triplet therapy in mCSPC patients is now under investigation in ongoing trials, such as the UpFront PSMA and PSMAddition [112]. Another critical aspect of prostate cancer treatment focuses on the phosphatidylinositol 3-kinase (PI3K)-AKT pathway, which plays a key role in tumorigenesis, progression, and treatment resistance [113]. In this area, the ProCAID trial assessed the combination of capivasertib, a selective inhibitor of all three AKT isoforms, with docetaxel chemotherapy [114]. The trial revealed a median OS benefit in the capivasertib + docetaxel arm, especially among patients who had previously been treated with abiraterone or enzalutamide. The ongoing CAPItello-281 study seeks to further investigate the potential of targeting the PI3K-AKT pathway, exploring the triplet combination of capivasertib, abiraterone, and ADT in PTEN-deficient mCSPC patients [115]. Meanwhile, advances in the understanding of immune checkpoint receptors and ligands have led to transformative breakthroughs in cancer treatment. Pembrolizumab, a PD-1 inhibitor, showed significant promise across various tumor types [116]. In the context of prostate cancer, the KEYNOTE-991 trial is currently evaluating the efficacy and safety of pembrolizumab in combination with enzalutamide and ADT in mCSPC patients compared to the standard regimen of enzalutamide + ADT [117]. This trial aims to determine whether the integration of immunotherapy can enhance treatment outcomes for these patients. 

The trend toward low adoption of combination treatment in real-world settings impedes the generation of efficacy data outside clinical trials. Additional therapeutic agents currently indicated for mCRPC are anticipated to be approved for earlier-stage hormone-sensitive disease as part of combination treatments with ADT [65,118,119]. This highlights the need for further investigation into the optimal application of combination therapies.

### 6.6. Treatment Intensification and De-Escalation

Intensifying treatment by combining multiple agents raises questions about therapy duration. Optimal application may involve determining how to intensify and de-escalate treatment [85]. Considering no established superiority of continuous over intermittent ADT, intermittent regimens should be considered after initial treatment intensification in selected patients [120]. PSA kinetics may serve as biomarkers to guide treatment de-escalation in mCSPC patients [7], although further trials are needed to define optimal strategies for timing and choice of agents.

Table 4 provides an overview of ongoing clinical trials and emerging therapeutic strategies in mCSPC.

## 7. Conclusions

The treatment landscape for mCSPC has evolved significantly, with studies showing that early treatment intensification using a combination of ADT, ARPIs, and chemotherapy improves survival. Despite strong evidence and guideline recommendations, real-world data indicate low adoption rates of these intensified treatments due to factors such as regulatory and reimbursement differences, resource limitations, and a lack of biomarkers for treatment selection. Increased awareness and efforts are needed to ensure evidence-based treatments are delivered to patients.

Combination therapy, including ADT with ARPIs and docetaxel, offers the best chance for long-term survival in eligible patients. Research continues to explore the molecular biology of mCSPC, treatment resistance mechanisms, and optimized schedules for targeted therapies. Clinical trials, such as PEACE-1 and ARASENS, demonstrated that adding ARPIs to ADT and docetaxel significantly improves survival, especially in patients with synchronous, HV mCSPC. However, a one-size-fits-all approach may lead to overtreatment. Furthermore, personalized treatment strategies must be based on factors such as disease volume, timing of metastases, and patient fitness (Figure 3).

Real-world data reveal that many mCSPC patients are still undertreated, often receiving only ADT without the recommended doublet or triplet therapies. This highlights a gap between clinical trial findings and their implementation in practice. Future clinical trials should continue to explore the optimal use of triplet therapy versus doublet combinations, particularly for different patient subgroups. Personalized treatment approaches that take into account disease characteristics and patient comorbidities are essential to improve outcomes and reduce the burden of overtreatment. As new evidence emerges, guidelines and practices must be updated to reflect the most effective treatment strategies.

## Figures and Tables

**Figure 2 cancers-16-03187-f002:**
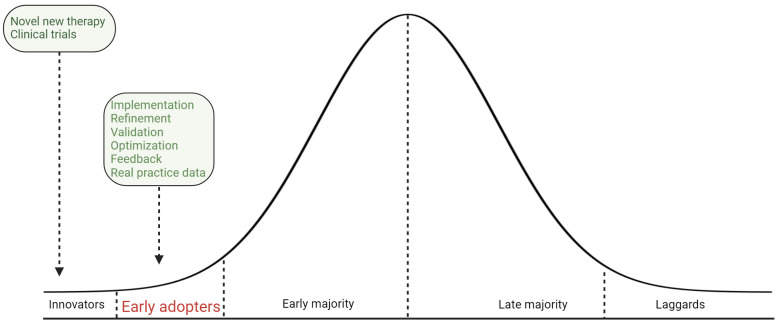
The adoption of combination therapy for mCSPC follows the diffusion of innovations theory over time. Early in the process, the uptake is primarily driven by “early adopters”, who are recognized as thought leaders in mCSPC treatment. As the adoption spreads, the majority of physicians gradually begin to incorporate the new treatment approach, represented by the expanding section of the violin plot that includes the “early majority” and “late majority”. Finally, the lower portion of the plot represents “laggards” or “late movers”, who are more resistant to change and take significantly longer to integrate this new clinical practice.

**Figure 3 cancers-16-03187-f003:**
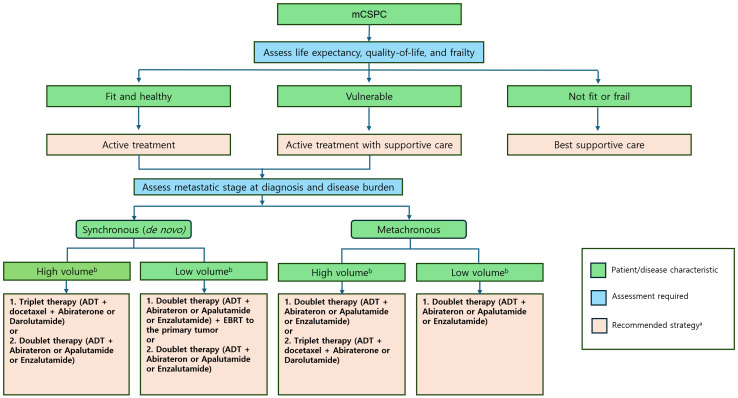
Recommended treatment approaches in patients with mCSPC. a. All treatment decisions should be made only after thoroughly discussing the benefits and risks with the patient and/or their caregiver. b. Disease volume was evaluated using the CHAARTED criteria [41]: high volume was characterized by the presence of visceral metastases and/or four or more bone metastases, with at least one metastasis located outside the vertebral column and pelvis. Abbreviations: ADT, androgen-deprivation therapy; mCSPC, metastatic castration-sensitive prostate cancer.

**Table 1 cancers-16-03187-t001:** Summary of key clinical trials evaluating combination doublet therapies in metastatic castrate-sensitive prostate cancer.

Trial	Patients Enrolled	Intervention Arm	Control Arm	Previous/Concurrent Docetaxel	Median Follow-Up (Months)	Median OS in the Intervention Arm (Months)	Median OS in the Control Arm (Months)	Hazard Ratio (HR) (95% CI)	*p*-Value	Most Common AEs (Any Grade; ≥15% of the Investigation Group) vs. the Comparator Group	Key Findings
STAMPEDE [44]	≈1000	ADT + abiraterone	ADT	Not allowed	40	79.0	46.0	HR: 0.61 (0.49–0.75)	<0.001	Hypertension (17% vs. 7%), hypokalemia (15% vs. 7%), fatigue (20% vs. 15%)	Significant OS improvement with abiraterone; benefits seen in both low- and high-risk groups
LATITUDE [46]	1199	ADT + abiraterone + prednisone	ADT + placebo	Not allowed	51.8	53.3	36.5	HR: 0.66 (0.56–0.78)	<0.001	Hypertension (20% vs. 10%), hypokalemia (18% vs. 9%), edema (16% vs. 10%)	Significant OS and rPFS improvement; notable but generally manageable AEs
TITAN [49]	1052	ADT + apalutamide	ADT + placebo	Allowed (11%)	44	Not reached	52.2	HR: 0.65 (0.53–0.79)	<0.001	Rash (27% vs. 8%), hypothyroidism (16% vs. 6%), ischemic heart disease (15% vs. 7%)	Substantial OS and rPFS benefit; consistent benefits across subgroups
ENZAMET [54]	1125	ADT + enzalutamide	ADT + first-generation antiandrogen	Allowed (concurrent 45%)	68	Not reported	Not reported	HR: 0.70 (0.58–0.84)	<0.001	Fatigue (22% vs. 14%), hypertension (16% vs. 10%), seizures (2% vs. 1%)	Significant OS and PFS improvements, particularly in synchronous disease
ARCHES [57]	1150	ADT + enzalutamide	ADT + placebo	Allowed (previous 18%)	44.6	Not reported	Not reported	HR: 0.66 (0.53–0.81)	<0.001	Fatigue (21% vs. 15%), hypertension (15% vs. 9%), seizures (1.5% vs. 0.5%)	Robust improvement in rPFS and OS; manageable safety profile
CHAARTED [61]	790	ADT + docetaxel	ADT	Not allowed	53.7	57.6	47.2	HR: 0.72 (0.59–0.89)	0.002	Neutropenia (40% vs. 10%), febrile neutropenia (16% vs. 5%), fatigue (18% vs. 10%)	Greater OS benefit in HV disease; limited benefit in LV disease
GETUG-AFU-15 [62]	385	ADT + docetaxel	ADT	Not allowed	83.9	Not significant	Not significant	HR: 1.01 (0.75–1.36)	0.14	Neutropenia (35% vs. 13%), febrile neutropenia (17% vs. 5%), fatigue (20% vs. 10%)	Trend toward OS benefit in HV disease; no significant OS benefit overall

Abbreviations: ADT, androgen deprivation therapy; OS, overall survival; HR, hazard ratio; HV, high volume; LV, low volume; rPFS, radiographic progression-free survival.

**Table 2 cancers-16-03187-t002:** Summary of key clinical trials evaluating combination triplet therapies in metastatic castrate-sensitive prostate cancer.

Trial Name	Patients Enrolled	Intervention Arm	Control Arm	% Synchronous	% High Volume	Median Follow-Up (Months)	Median OS in the Intervention Arm (Months)	Median OS in the Control Arm (Months)	Group: HR (95% CI)	Key Adverse Events	Key Findings
PEACE-1 [73]	1173	SOC + abiraterone (with or without RT)	SOC (with or without RT)	100%	64%	45.6	NR	53.2	0.75 (0.59–0.95)	Hypertension, transaminase increase	Significant improvement in OS and rPFS, particularly in HV disease. Higher incidence of grade ≥ 3 AEs in triplet therapy.
ARASENS [74]	1306	ADT + docetaxel + darolutamide	ADT + docetaxel + placebo	86%	77%	43.7	NR	48.9	0.68 (0.57–0.80)	Rash, hypertension	Improved OS and secondary endpoints, consistent benefit across subgroups. Less clear benefit in LV disease.
ENZAMET [55]	1125	ADT + docetaxel + enzalutamide	ADT + docetaxel + first-generation antiandrogen	72%	71%	68	Not reported	Not reported	0.73 (0.55–0.99)	Similar to previous trials with enzalutamide	Significant OS improvement in synchronous mCSPC, not in metachronous disease. Early chemotherapy beneficial in high-risk patients.

Abbreviations: ADT, androgen deprivation therapy; AE, adverse event; HV, high volume; LV, low volume; mCSPC, metastatic castrate-sensitive prostate cancer; OS, overall survival; rPFS, radiographic progression-free survival; SOC, standard of care.

**Table 3 cancers-16-03187-t003:** Summary of key network meta-analyses evaluating combination therapies in metastatic castrate-sensitive prostate cancer.

Study	Focus	Treatment Comparisons	Key Findings (LV mCSPC)	Key Findings (HV mCSPC)	Other Notes
Hoeh et al. (2023) [76]	Comparative efficacy of triplet vs. doublet therapies in mCSPC stratified by disease volume	ARPI + ADT or docetaxel + ADT vs. ARPI + docetaxel + ADT	No significant OS differences between triplet and doublet therapies; other combinations did not show benefits over ADT alone	All combinations improved OS compared to ADT alone; darolutamide + docetaxel + ADT ranked the highest in OS	Emphasized the importance of stratifying patients by disease volume for treatment decisions
Jian et al. (2023) [78]	Systematic review and NMA comparing the efficacy of combination therapies in mCSPC	Various combination therapies, including triplet and doublet regimens	ADT + ARAT ranked the highest in OS and rPFS; triplet therapies showed no OS or rPFS improvements and a higher risk of AEs	Triplet therapy ranked first in OS and rPFS; ADT + rezvilutamide and ADT + docetaxel were also effective	Highlighted the need for careful consideration of disease volume due to the increased risk of AEs with triplet therapies
Riaz et al. (2023) [77]	Evaluation of systemic treatment options for mCSPC	Triplet therapies vs. doublet therapies	Triplet therapies did not significantly outperform ARPI doublets or docetaxel + ADT; higher risk of AEs	Triplet therapies showed OS advantage; darolutamide and abiraterone triplets significantly improved OS over docetaxel + ADT	Subgroup analyses showed that triplet therapies provided OS advantage for HV disease but not LV; emphasized safety vs. efficacy balance

**Table 4 cancers-16-03187-t004:** Summary of ongoing issues and related trials in metastatic castrate-sensitive prostate cancer.

Topic	Details	Related Ongoing Trials
Optimal agent following triplet therapy	Cabazitaxel: ability to overcome taxane resistance and manageable toxicity profile LuPSMA: targeted approach for PSMA-positive disease PARPis: potential benefits in HRRm patients Each agent offers distinct advantages depending on the patient’s disease characteristics and prior treatment history.	CASCARA: assessing quadruplet therapy (ADT + cabazitaxel/carboplatin + abiraterone) in high-volume mCSPC [103] UpFront PSMA: evaluating LuPSMA in mCSPC [104] AMPLITUDE: assessing efficacy of PARPis in mCSPC with HRRm [105]
Post-triplet therapy strategy for HRRm patients	PARPi monotherapy: leveraging synthetic lethality in HRR-deficient cells with high efficacy and manageable safety profile PARPis and ARPIs combination: potential synergistic effects but increased risk of compounded toxicities Early data from TALAPRO-3 suggest potential benefits.	TALAPRO-3: assessing combination therapy of PARPi and ARPi in mCSPC [121]
Comparative efficacy of triplet therapy vs. PARPis in the HRRm population	Active research comparing triplet therapy and PARPis TALAPRO-3 and AMPLITUDE trials expected to provide critical insights Preliminary data suggest significant advantages of PARPis in HRRm patients.	AMPLITUDE and TALAPRO-3: expected to provide critical insights [105,121]
Need for head-to-head (combination of ADT with ARPIs vs. triplet combination of ADT, docetaxel, and ARPIs)	Limited direct comparisons currently available Further research is needed to directly compare ADT + ARPIs vs. triplet therapy.	PEACE-1: comparing SOC (ADT alone or with docetaxel) vs. SOC plus abiraterone in mCSPC [73] ARASENS: comparing ADT with docetaxel + placebo vs. ADT with docetaxel + darolutamide in mCSPC [74]
Novel emerging therapeutic agents	LuPSMA: promising results in VISION and TheraP trials Capivasertib: targeting the PI3K-AKT pathway Pembrolizumab: evaluating the potential role in immunotherapy Emerging therapies are likely to be approved for earlier-stage CSPC.	UpFront PSMA: evaluating LuPSMA in mCSPC [104] PSMAddition: comparing SOC (ADT with ARPI) vs. SOC plus ^177^Lu-PSMA-617 in mCSPC [122] CAPItello-281: assessing capivasertib, abiraterone, and ADT in PTEN-deficient mCSPC [115] KEYNOTE-991: evaluating pembrolizumab, enzalutamide, and ADT in mCSPC [117]
Treatment intensification and de-escalation	Combining multiple agents raises questions about the duration of therapy Intermittent regimens should be considered after initial treatment intensification in selected patients PSA kinetics may serve as biomarkers to guide treatment de-escalation; further trials are needed.	STAMPEDE: evaluating efficacy of ADT with various drug combinations [123] SWOG S1216: evaluating treatment intensification and de-escalation strategies based on PSA response [124]

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
