# Peer review of "Strategic Advances in Combination Therapy for Metastatic Castration-Sensitive Prostate Cancer: Current Insights and Future Perspectives"

_cancers, 2024, doi:10.3390/cancers16183187_

Round 1
Reviewer 1 Report
Comments and Suggestions for Authors
Title:
Strategic Advances in Combination Therapy for Metastatic Cas- 2 tration-Sensitive Prostate Cancer: Current Insights and Future 3 Perspective
The treatment strategy of patients with metastatic castration-sensitive prostate cancer has changed considerably in recent years, with a shift from monotherapy to intensification with doublet or triplet therapies. This has made it possible to investigate the use of new molecules in combination or alone, and to reduce the risk of tumour resistance and thus improve failure-free survival, OS and QoL.
Highlights of the study:
1 Current topic of interest
2 The review done by the authors was done with correct method and comprehensively incorporates the recommendations of international guidelines and recent publications.
3The tables have been well constructed, to facilitate the interpretation for the user.
Weaknesses of the study:
Not detected
Request for revision:
Accept in present form
Author Response
Thank you for your positive feedback and for accepting our manuscript titled "Strategic Advances in Combination Therapy for Metastatic Castration-Sensitive Prostate Cancer: Current Insights and Future Perspective." We are grateful for the thorough review and appreciate the constructive comments provided.
We are delighted that the manuscript meets the journal's standards in its current form. We would like to express our gratitude to you and the reviewers for your time and effort in evaluating our work.
Reviewer 2 Report
Comments and Suggestions for Authors
The topic is interesting, and this is very timely and comprehensive review of the important clinical and scientific subject. Manuscript is a well written analysis and well organized of a large number of clinical studies. Overall, i find it interesting, so can be published.
Author Response

(The authors gave the same response as above.)

Reviewer 3 Report
Comments and Suggestions for Authors
Dear authors,
The submitted manuscript effectively summarizes and aims to clarify the complex topic of treatment strategies for metastatic castration-resistant prostate cancer (mCRPC). The objective of the study is well met, providing a comprehensive and clear overview of the current therapeutic approaches. The manuscript is well-written and thorough, making the content accessible even to those who may not be directly involved in the field. The figures used, particularly the summary figure, offer a visually effective representation, providing a clear graphic idea of the current landscape.
Overall, I believe the manuscript is well-structured, the data supports the conclusions, and it is suitable for publication.
Author Response

(The authors gave the same response as above.)
